

# Function, structure and quality of striated muscles in the lower extremities in patients with late onset Pompe Disease—an MRI study

Michael Vaeggemose[1], Rosa Andersen Mencagli[1], Julie Schjødtz Hansen[1], Bianca Dräger[2], Steffen Ringgaard[3], John Vissing[4] and Henning Andersen[1]

[1] Department of Neurology, Aarhus University Hospital, Aarhus N, Denmark
[2] Department of Sleep Medicine and Neuromuscular Disorders, University Hospital Muenster, Münster, Germany
[3] MR Research Centre, Aarhus University, Aarhus N, Denmark
[4] Copenhagen Neuromuscular Center, Department of Neurology, Rigshospitalet, University of Copenhagen, Copenhagen, Denmark

## ABSTRACT

**Background**. Pompe Disease (PD) is a rare inherited metabolic myopathy, caused by lysosomal-$\alpha$-glucosidase (GAA) deficiency, which leads to glycogen accumulation within the lysosomes, resulting in cellular and tissue damage. Due to the emergence of a disease modifying treatment with recombinant GAA there has been a large increase in studies of late onset Pompe Disease (LOPD) during the last decade.

**Methods**. The present study evaluates muscle quality in 10 patients with LOPD receiving treatment with enzyme replacement therapy and in 10 age and gender matched healthy controls applying $T_1$-weighted Dixon MR imaging and isokinetic dynamometry. Muscle quality was determined by muscle strength in relation to muscle size (contractile cross-sectional area, CSA) and to muscle quality (fat fraction). A follow-up evaluation of the patients was performed after 8–12 months. Patient evaluations also included: six-minute walking test (6MWT), forced vital capacity, manual muscle testing and SF-36 questionnaire.

**Results**. Fat fraction of knee flexors (0.15 vs 0.07, $p < 0.05$) and hip muscles (0.11 vs 0.07, $p < 0.05$) were higher in patients than controls. In patients, contractile CSA correlated with muscle strength (knee flexors: $r = 0.86$, knee extensors: $r = 0.88$, hip extensors: $r = 0.83$, $p < 0.05$). No correlation was found between fat fraction and muscle strength. The fat fraction of thigh muscles did not correlate with scores from the clinical tests nor did it correlate with the 6MWT. During follow-up, the contractile CSA of the knee extensors increased by 2%. No other statistically significant change was observed. Quantitative MRI reflects muscle function in patients with LOPD, but larger long-term studies are needed to evaluate its utility in detecting changes over time.

Corresponding author
Michael Vaeggemose,
mivaeg@clin.au.dk

## INTRODUCTION

Pompe Disease (PD) is a rare inherited metabolic myopathy, caused by lysosomal-$\alpha$-glucosidase (GAA) deficiency, which leads to glycogen accumulation within the lysosomes, resulting in cellular and tissue damage (*Hirschhorn & Reuser, 2001*). PD presents clinically in two forms, the infantile Pompe with no residual enzyme activity (*Kishnani et al., 2006*; *Van Der Ploeg & Reuser, 2008*), and the less severe late onset Pompe Disease (LOPD). LOPD may be symptomatic from young age to adulthood and is characterized by some residual activity of GAA. The skeletal muscles gradually degenerate with accumulating fat infiltration (*Carlier et al., 2011*). Skeletal muscle dysfunction results in muscle weakness, fatigue and respiratory insufficiency. With disease progression, patients may become wheelchair-bound and need ventilator support (*Van Der Ploeg & Reuser, 2008*; *Hagemans et al., 2005*), and have reduced life expectancy (*Güngör et al., 2011*).

Due to the emergence of a disease modifying treatment with recombinant GAA there has been a large increase in studies of LOPD during the last decade, which have expanded our knowledge about LOPD as a broader phenotype of a multisystemic disease (*Chan et al., 2017*). Treatment with recombinant GAA, also known as enzyme replacement therapy (ERT), has been available for LOPD since 2006, and new drugs are under development. Enzyme replacement therapy (ERT) has a well-established but moderate effect in most patients, however, the response to the treatment varies (*Van der Ploeg et al., 2017*). In a slowly progressing disease such as LOPD, monitoring the effect of a treatment is challenging, particularly as the treatment merely attenuates the disease progression rather than improving the functional capacity. The European Guidelines (*Schoser et al., 2015*) recommend the following three endpoints for monitoring the disease: I. Functional tests; primarily the 6 min walking test (6MWT) and the four timed tests (walking 10 metres, climbing four steps, standing up from supine position and standing up from a chair); II. Forced vital capacity (FVC) in sitting and lying position and maximal inspiratory/expiratory pressure (MIP/MEP); III. Muscle strength assessed by manual muscle test (MRC).

However, the recommended tests are susceptible to variation in performance, due to patient dependence (lack of motivation and concentration, fatigue, pain, learning effects, and inherent variations in the measurements). This impedes detection of small changes in the patient's performance, especially over short periods of time and in small patient populations. Thus, there is still the need for sensitive and reliable objective methods to monitor LOPD patients.

Magnetic resonance imaging (MRI) is a promising tool as it enables identification and quantification of fat replacement in skeletal muscles. In recent years, there has been a growing interest in the clinical use of MRI of skeletal muscles in neuromuscular disorders (*Kalia et al., 2017*; *Carlier et al., 2016*; *Burakiewicz et al., 2017*; *Fischer, Bonati & Wattjes, 2016*). Studies have reported close correlations between MRI findings and clinical tests although semi quantitative methods were primarily used. Interestingly, quantitative MRI has also been shown to enable detection of subclinical changes in fat fraction over time (*Carlier et al., 2015*). However, results are difficult to compare as some MRI studies have not included quantitative techniques to determine strength of individual muscle

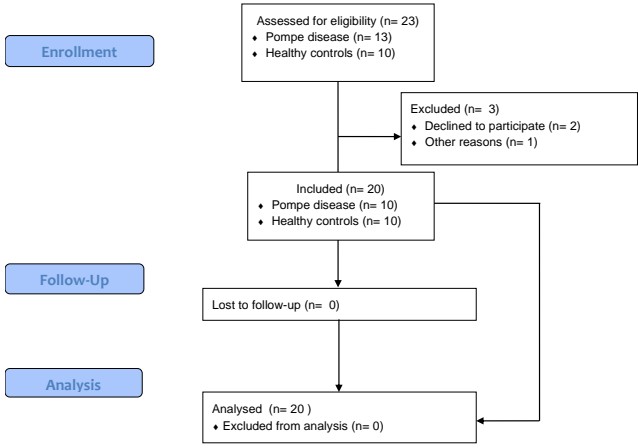

**Figure 1**  **Flow chart of late onset pompe disease and healthy control subjects participating in the study.**

groups (*Khan et al., 2019*), have only performed cross sectional studies (*Figueroa-Bonaparte et al., 2016*) or have only included a few muscle groups (*Rehmann et al., 2020*).

Therefore, the aims of this study were to evaluate the quality of a high number of muscles in patients with LOPD by determining the correlation between isokinetic muscle strength and the contractile cross-sectional area (CCSA). Furthermore, to evaluate whether changes in fat infiltration and the contractile cross sectional determined by MRI in the lower extremity corresponds to changes in disease severity during a shorter follow up period.

# MATERIALS AND METHODS

## Population

Between September 2015 and May 2017, all 11 Danish patients diagnosed with LOPD were invited to participate in the study. Eight accepted participation, one did not answer, and two declined participation due to the need for respiratory support during the MR scan. To increase the statistical power of the study, two patients were recruited from the Department of Sleep Medicine and Neuromuscular Disorders at Münster University Hospital, Germany, resulting in a total of 10 patients with a genetically confirmed diagnosis of LOPD (Fig. 1). All patients were examined at day 1 and again after 8 months, except 3 patients, who were diagnosed during the study period. They were examined after 12 months to experience a complete 8 months of drug administration for study comparison. Nine patients received alglucosidase alfa and 1 patient received avalglucosidase alfa (neoGAA).

For comparison, 10 healthy controls matched for age, sex, height and weight were recruited and examined. Controls were recruited through advertisement in a local newspaper, on an online webpage for healthy controls and through public posters.

Exclusion criteria where: Age <18 years, contraindications to MRI (metal in the body, pacemaker, pregnancy, and claustrophobia), cancer and other serious medical conditions and abuse of alcohol. Healthy control subjects were only examined by isokinetic dynamometry and MRI. All participants gave informed signed consent and were examined
at the Department of Neurology, Aarhus University Hospital, Denmark. The study was approved by the Danish National Committee On Health Research Ethics (application no. 49089) and by the Danish Data Protection Agency. The study was registered at ClinicalTrials.gov (identifier: NCT02708784).

## MRI acquisition

MR scanning was performed at the MR Centre at Aarhus University Hospital using a 3-T Siemens Skyra Magnetom (Siemens AG, Erlangen, Germany). The scan protocol consisted of a T1-weighted, 3-point gradient echo Dixon sequence (400 slices of three mm thickness, 20% gap, TR 5.31 ms, TE 2.46 ms, flip angle 9°, FOV 500, in-plane resolution 1 x 1 mm) acquiring images in a total scan time of 40 min. Participants were lying in a feet-first supine position and MR images were recorded from the ankle (lateral malleolus) to the trochanter major with the use of a 36 channel peripheral angio coil (Siemens AG, Erlangen, Germany) and from the trochanter major to L1 with a 18 channel body matrix coil (Siemens AG, Erlangen, Germany).

## MRI data analysis

MR images were analysed using Osirix (Version 6.5.2, Pixmeo SARL, Switzerland). Muscle segmentation was done manually by segmenting regions of interest (ROI) in the T1-weigthed opposed-phase images, recorded with the Dixon sequence. The following 12 muscles were segmented at the non-dominant thigh: biceps femoris caput longum muscle, semimembranosus muscle, semitendinosus muscle, rectus femoris muscle, vastus medialis muscle, vastus lateralis muscle, vastus intermedius muscle, sartorius muscle, gluteus maximus muscle, adductor magnus muscle, adductor longus muscle and iliopsoas muscle. To determine the muscle and fat fractions the ROIs were applied to the corresponding fat and water images calculated from the Dixon sequence. Based on the signal intensity (SI) the muscle fat fraction was calculated as the signal intensity from the fat image divided with the combined fat and water signal intensity from a given ROI (FF = SI(fat) / [SI(fat) + SI(water)]).

The fat fraction of the knee flexors was calculated as the mean fat faction from the following muscles: biceps femoris caput longum, semimembranosus and semitendinosus; the fat fraction of the knee extensors was calculated including the muscles: sartorius, rectus femoris, vastus lateralis, vastus intermedius and vastus medialis.

The contractile cross-sectional area (CSA) was determined as the cross-sectional muscle area subtracted the fat infiltrated muscle area (CSA = Area * [1 - FF]) similar to the method described by *Carlier et al. (2016)*.

To reduce bias in segmentation of muscles close to ligaments five cross sections of each muscle were segmented with respect to the centre slice between two bone fix points illustrated by Fig. 2. For the knee extensors, the knee flexors, the adductor magnus muscle and the adductor longus muscle, the slices chosen corresponded to the middle five slices between the trochanter major on the femur and the upper edge of the patella. The five slices analysed to evaluate the gluteus maximus muscle and the iliopsoas muscle corresponded to the most proximal part of the femur bone. The slices chosen correspond to the slice where

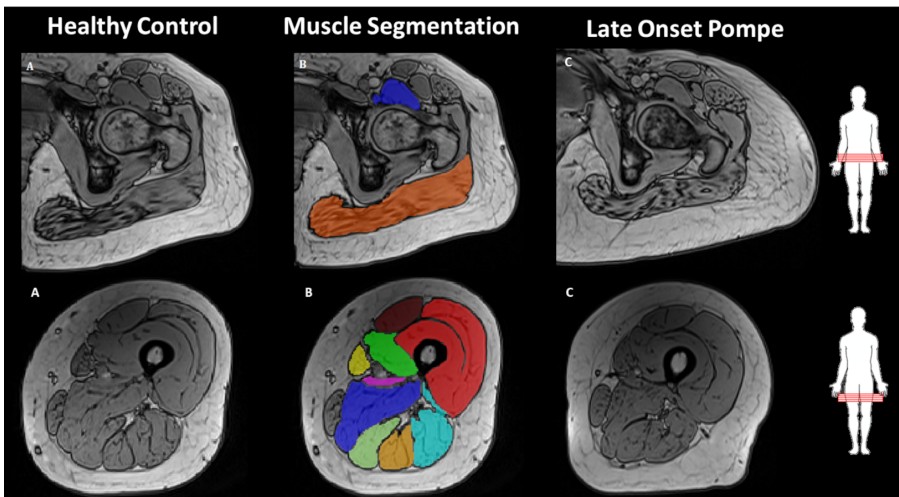

**Figure 2** Illustration of muscle segmentation in the opposed phase Dixon MR images of the muscles of hip flexor/extensor and knee flexors/extensors, corresponding to the following individual muscles. *M. iliopsoas, M. gluteus maximus, M. rectus femoris, M. sartorius, M. vastus medialis, M. vastus intermedius, M. vastus lateralis, M. biceps femoris, M. semimembranosus, M. semitendinosus.* The images are shown at comparable slices of (A) a healthy subject, (B) segmentation of the healthy subject muscles', and (C) a late onset Pompe subject.

the caput femoris first appears together with the two nearby proximal and distal slices. Although, this results in segmentation of the most distal part of the iliopsoas muscle, which is not representative for the whole muscle, this segmentation was chosen as respiration and bowel artefacts was present in images of the *m. iliopsoas* more proximally. The vastus intermedius and the vastus lateralis muscles are difficult to discriminate and therefore the muscles were analysed together, as proposed by *Barnouin et al. (2014)*. The short head on the biceps femoris muscle was not analysed, as the muscle was not visible in all subjects on the chosen slices. Due to respiratory difficulties patient 1 was supported with cushions enabling only MRI images from the ankle to the upper knee.

## Isokinetic muscle strength

Isokinetic muscle strength was assessed by dynamometry using a Biodex System 3 PRO dynamometer (Biodex Medical System, NY USA; software version 3.1) as described earlier (*Harbo, Brincks & Andersen, 2012*). The protocol included eight repetitions for each movement and standardized instructions before and during the tests. The maximal peak torque was used for further analysis together with the percentage of expected strength, calculated according to Harbo et al. In case of a coefficient of variation >15%, the test was repeated. Maximal muscle strength was determined for the flexors and extensors of the ankle, knee, hip, elbow and wrist, together with shoulder abduction and adduction. Patient 1 was unable to perform hip flexion and extension, due to need of respiratory support in supine position. Patients 3, 6 and 9 were not able to perform all the tests due to muscle weakness and pain. Patient 3 could only perform hip flexion, shoulder abduction, elbow flexion and hand flexion and extension. Patient 6 was unable to perform hip flexion and

**Table 1  Subjects' demographics.**

|  | LOPD | Controls |
|---|---|---|
| Number of subjects | 10 | 10 |
| Female (%) | 5 (50) | 5 (50) |
| Age (years) | 36 (19-62) | 38 (21-62) |
| Height (cm) | 178 (161–193) | 174 (161–180) |
| Weight (kg) | 72.4 (58–102) | 72.9 (59–92) |

Notes.
  Values are given in mean values and range.

extension. Patient 9 was unable to perform hip flexion, shoulder abduction and elbow flexion.

## Clinical measurements

Muscle strength was determined by manual muscle testing by a trained neurologist (HA) and graded as follows (0 = full strength, 1 = 25% reduced strength, 2 = 50% reduced strength, 3 = 75% reduced strength and 4 = 100% reduced strength). Exercise capacity was assessed by the 6 min walking test (6MWT). Vital capacity was determined according to the European Respiratory Society Guidelines (*Quanjer et al., 1993*) using a spirometry system in both sitting and supine position. The highest measurement out of three consecutive tests, defined as % of expected forced vital capacity (FVC) in litres, was used in further analyses. Perceived quality of life was established according to the Danish version of the 36-Item Short Form (SF-36) questionnaire.

Finally, the patients were ranked according to each of these clinical measurements, and a summed rank-score was obtained as a measure of the combined disease severity status based on the 10 participating LOPD subjects.

## Statistical analysis

Muscle strength, fat fraction and contractile CSA in LOPD patients were compared to healthy controls by paired *t*-test and Wilcoxon signed rank test. Correlations between muscle strength and muscle quality and between total fat fraction and clinical tests were tested by Spearman's correlation and linear regression both for patients and controls. Follow-up data were analysed using paired *t*-test. Statistical significance was determined with a *p*-value of 0.05 or less.

# RESULTS

Demographics of the ten patients with LOPD and ten healthy matched controls are shown in Table 1. Furthermore, the clinical characteristics with gene mutations of the patient population are described in Table 2.

## Contractile cross-sectional area and fat fraction

Contractile CSA of the knee flexors, the knee extensors, the gluteus maximus muscle and the adductor magnus muscle was reduced in the patients compared to their healthy controls (Table 3). However, no statistical difference was found for the iliopsoas muscle and the
**Table 2  Clinical characteristics of the LOPD patients.** During the study period patient 1-9 received alglucosidase alfa.

| Patient | Sex | Age | Year of diagnosis | Start on ERT | Aids | Gene mutation |
|---------|-----|-----|-------------------|--------------|------|---------------|
| 1 | M | 62 | 2000 | 2007 | B-pap at night | GAA-mutation: c*-32-13T>A<br>GAA-mutation: c.1003G>A |
| 2 | F | 43 | 2010 | 2011 | ÷ | GAA-mutation: c-32-13T>A<br>GAA-mutation: c.307T>G;<br>pC103G |
| 3 | F | 40 | 2015 | 2015 | ÷ | GAA-mutation: c.32-13T>G,<br>Homozygous |
| 4 | M | 42 | 2015 | 2015 | ÷ | GAA-mutation: c.32-13T>G<br>GAA-mutation: c.525 del,<br>p. Glu176Argfs*45 |
| 5 | M | 44 | 2010 | 2011 | ÷ | GAA-mutation: c.32-13T>A<br>GAA-mutation: c.2331+2T>G |
| 6 | F | 30 | 2016 | 2016 | Wheel chair for longer distance | GAA-mutation: c-32-13T>G<br>GAA-mutation: c-525 del,<br>p. Glu176Argfs*45 |
| 7 | F | 25 | 2013 | 2014 | Wheel chair for longer distance | GAA-mutation: c-32-13T>G<br>GAA-mutation: c-525 del,<br>p. Glu176Argfs*45 |
| 8 | M | 19 | 2014 | 2015 | ÷ | GAA-mutation: c.32-13T>G<br>GAA-mutation: c.1548G>A |
| 9 | F | 34 | 2003 | 2014 | ÷ | GAA-mutation: c.32-13T>G<br>GAA-mutation: c.1802C>T |
| 10 | M | 22 | 1995 | 2006 | ÷ | GAA-mutation: IVS1-13T >G<br>GAA-mutation: 2.228A>G |

**Notes.**

During the study period patient 1–9 received alglucosidase alfa. Patient 10 received avalglucosidase alfa (neoGAA).

ERT, Enzyme replacement therapy.

adductor longus muscle. In all muscles examined, patients had a higher fat fraction, apart from the knee extensors (Table 3 and Fig. 3). In the lower limbs, patients had reduced strength of knee extensors, hip flexors and hip extensors. In the upper limb, strength was reduced for shoulder adductors and elbow flexors only (Table 3).

### Muscle quality

Close correlations were found between the contractile CSA and muscle strength of the knee extensors in both healthy controls and patients (Table 4).

In patients, the contractile CSA of the knee flexors and hip extensors and the fat fraction of the knee flexors also correlated with muscle strength.

No correlation was found between the total fat fraction and the summed rank-score determined from the clinical tests, nor between the total fat fraction and the 6MWT (Table 4).

### Longitudinal analysis

During the follow-up period no change was observed at any of the clinical tests (Table 5).

Contractile CSA of the knee extensors increased slightly. The other contractile CSA and the fat fractions remained stable (Tables 6 and 7).

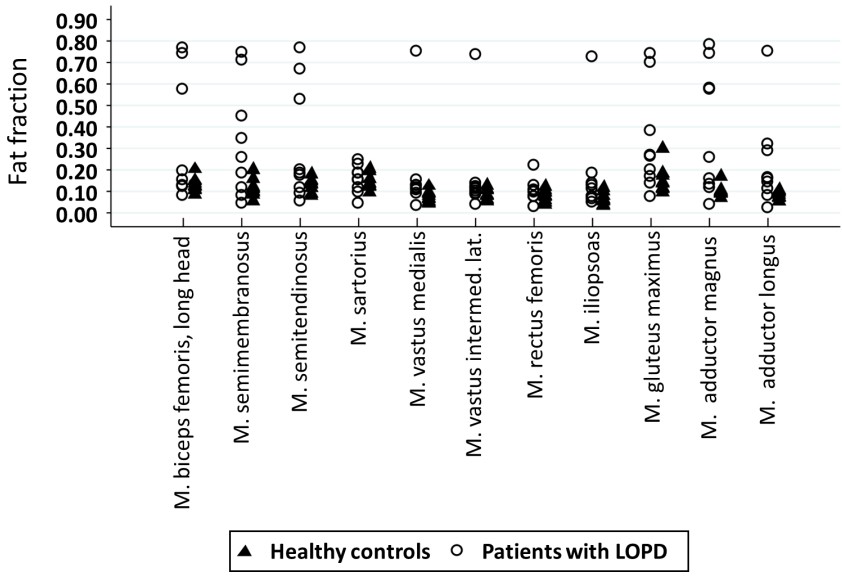

**Figure 3** Dot plot illustrating the fat fraction of each muscle in patients (circles) and controls (triangles).

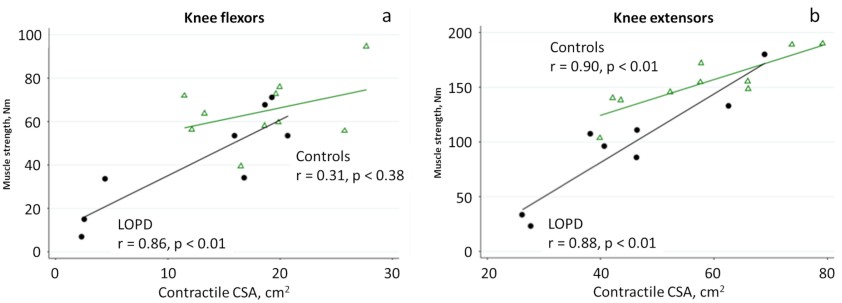

**Figure 4** Linear regression analysis of muscle strength and contractile CSA of the knee flexors (A) and knee extensors (B) in patients with LOPD (circle) and controls (triangle).

No correlation was found between the change in expected muscle strength and the change in the total fat fraction, neither between the relative changes in % of total expected muscle strength and total fat fraction or total CSA (Fig. 4).

## DISCUSSION

Fat fraction determined by MRI was higher in patients with LOPD as compared to healthy controls. The replacement of muscle by fat followed a selective pattern, with more involvement of the adductor magnus muscle, the muscles at the pelvic girdle and the hamstrings combined with selective sparing of the short head of the biceps femoris muscle. This finding is in line with previous studies (*Carlier et al., 2011*; *Pichiecchio et al., 2004*). In patients with more advanced disease there was fat replacement in knee extensors also,

**Table 3   Fat fraction, contractile CSA and muscle strength in patients and in healthy controls.**

| | | Patients | Controls |
|---|---|---|---|
| Fat fraction | Knee-extensors[1] | 0.12 (0.03–0.49) | 0.09 ± 0.02 |
| | *M. sartorius*[1] | 0.16 ± 0.06 | 0.14 ± 0.03 |
| | *M. rectus femoris*[1] | 0.10 ± 0.05 | 0.08 ± 0.03 |
| | *M. vastus medialis*[1] | 0.11 (0.03–0.75) | 0.07 ± 0.02 |
| | *M. vastus lateralis. et intermedius* [1] | 0.11 (0.04–0.73) | 0.08 ± 0.03 |
| | **Knee-flexors[1]** | **0.21 (0.06-0.75)**[*] | **0.12 ± 0.03** |
| | **M. biceps femoris, caput longum[1]** | **0.15 (0.08-0.76)** | **0.12 ± 0.02** |
| | **M. semimembranosus[1]** | **0.32 ± 0.26**[*] | **0.11 ± 0.04** |
| | **M. semitendinosus[1]** | **0.18 (0.05-0.77)**[*] | **0.11 ± 0.03** |
| | **M. adductor magnus[1]** | **0.37 ± 0.29**[*] | **0.10 ± 0.03** |
| | **M. adductor longus[1]** | **0.16 (0.02–0.75)**[*] | **0.08 ± 0.02** |
| | **M. gluteus maximus[1]** | **0.32 ± 0.24**[*] | **0.15 ± 0.06** |
| | **M. iliopsoas[1]** | **0.11 (0.05–0.73)**[*] | **0.07 ± 0.03** |
| Contractile CSA (cm$^2$) | **Knee-extensors[1]** | **45.53 ± 14.38**[*] | **57.84 ± 15.51** |
| | *M. sartorius*[1] | 2.81 ± 1.10 | 2.54 ± 0.67 |
| | *M. rectus femoris*[1] | 7.15 ± 2.01 | 6.02 ± 1.76 |
| | **M. vastus medialis[1]** | **4.67 ± 2.66**[**] | **9.26 ± 2.88** |
| | **M. vastus lateralis et intermedius[1]** | **30.79 ± 12.05**[*] | **40.61 ± 9.41** |
| | **Knee-flexors[1]** | **16.83 (2.34-20.70)**[*] | **18.49 ± 5.41** |
| | **M. biceps femoris, caput longum[1]** | **6.56 (0.66–8.49)**[*] | **7.66 ± 2.56** |
| | **M. semimembranosus[1]** | **2.19 ± 1.57**[*] | **2.64 ± 2.17** |
| | *M. semitendinosus*[1] | 5.54 ± 3.48 | 7.03 ± 1.84 |
| | **M. adductor magnus[1]** | **7.87 ± 6.83**[**] | **20.72 ± 5.02** |
| | *M. adductor longus*[1] | 3.11 ± 1.98 | 4.56 ± 1.36 |
| | **M. gluteus maximus[1]** | **28.50 ± 14.76**[*] | **39.49 ± 8.43** |
| | *M. iliopsoas*[1] | 5.98 ± 3.28 | 6.68 ± 1.27 |
| Muscle strength (Nm) | **Knee-extensors[1]** | **103.2 ± 52.3**[**] | **153.4 ± 25.6** |
| | Knee-flexors[1] | 46.0 ± 25.2 | 64.5 ± 12.6 |
| | **Hip extensors[2]** | **62.8 ± 42.3**[*] | **139.8 ± 25.5** |
| | **Hip flexors[4]** | **77.0 ± 35.1**[**] | **120.6 ± 29.7** |
| | Foot flexors[1] | 52.4 ± 13.5 | 62.0 ± 11.3 |
| | Foot extensors[1] | 20.3 ± 15.2 | 23.4 ± 14.7 |
| | Shoulder abductor[1] | 37.8 ± 15.7 | 44.8 ± 12.9 |
| | **Shoulder adductor[1]** | **43.4 ± 23.1**[*] | **62.1 ± 19.2** |
| | **Elbow flexors[1]** | **33.6 ± 11.7**[*] | **40.6 ± 14.2** |
| | Elbow extensors[1] | 33.2 ± 11.5 | 35.8 ± 8.4 |
| | Hand flexors[1] | 16.6 ± 7.1 | 14.9 ± 4.1 |
| | Hand extensors | 8.8 ± 4.2 | 9.6 ± 1.7 |

**Notes.**
[*] $p < 0.05$.
[**] $p < 0.01$.
Values are given as mean and standard deviation or median and range. The superscript indicates the number of missing data in the patients group in the given analysis, e.g., fat-fraction [1] indicates one missing data. Statistically significant differences are presented in bold.

**Table 4 Correlations between muscle strength and the corresponding MRI findings from the muscles (contractile CSA and fatfraction) and between the fatfraction and the clinical tests (sum rank score and 6MWT).**

|  |  | Patients | Controls |
|---|---|---|---|
| Contractile CSA | Muscle strength |  |  |
|  | **Knee flexors[2]** | **$r = 0.86^{**}$** | $r = 0.31$ |
|  | **Knee extensors[2]** | **$r = 0.88^{**}$** | $r = 0.90^{**}$ |
|  | **Hip extensor[2]** | **$r = 0.83^{*}$** | $r = 0.38$ |
|  | Hip flexor[3] | $r = 0.29$ | **$r = 0.67^{**}$** |
| Fat fraction | Muscle strength |  |  |
|  | **Knee flexors[2]** | **$r = -0.83^{*}$** | $r = -0.14$ |
|  | Knee extensors[2] | $r = -0.43$ | $r = -0.25$ |
|  | Hip extensor[2] | $r = -0.68$ | $r = 0.07$ |
|  | Hip flexor[3] | $r = -0.14$ | $r = 0.24$ |
| Total fat fraction | Sum rank-score[1] | $r = 0.38$ |  |
| Total fat fraction | 6MWT[1] | $r = -0.50$ |  |

**Notes.**
$^{*}p < 0.05.$
$^{**}p < 0.01.$
Analysis was done by Spearman's correlation. The superscript indicates the number of missing data in the patients group in the given analysis, e.g. knee flexors 2 indicates missing data from two patients.

**Table 5 Clinical test at baseline and at follow-up.** MMT = Manual muscle testing.

|  | Baseline | Follow-up |
|---|---|---|
| 6MWT (m) | $506.6 \pm 143.89$ | $490.3 \pm 164.38$ |
| % of expected FVC |  |  |
| sitting position | $88 \pm 28$ | $87 \pm 24$ |
| lying position | $75 \pm 33$ | $75 \pm 29$ |
| MMT | $17.2 \pm 12.28$ | $15.2 \pm 10.99$ |
| SF-36 | $532.98 \pm 168.28$ | $464.07 \pm 175.37$ |
| % of expected total strength | $63 \pm 13$ | $63 \pm 11$ |

**Notes.**
MMT, Manual muscle testing; SF-36, = 36-item short form questionnaire. 6 minute walking test (6MWT) is measured in minutes.

however, there was selective sparing of the rectus femoris, sartorius and gracilis muscles. Although the fat fraction of the healthy controls was lower than the patients', the controls had a surprisingly high fat fraction compared to earlier studies (*Hogrel et al., 2015*; *Morrow et al., 2016*; *Horvath et al., 2015*). This could be due to inclusion of physically less active subjects as in earlier studies.

Contractile CSA of the knee extensors closely correlated with isokinetic strength both in patients and controls. In patients, close correlations were also found between the contractile CSA of the knee flexors and hip extensors when compared to muscle strength. However, no correlation was found for hip flexion, which may due to a smaller muscle size. The iliopsoas muscle was analyzed only very distally due to bowel and respiration artefacts, which may explain the lack of correlation between contractile CSA and muscle strength. *Pichiecchio et al. (2004)* reported close correlations between muscle volume and strength, however,

**Table 6  The contractile cross-sectional area (CSA) at baseline and at follow-up.**

|  | Baseline | Follow-up | Mean difference + SD |
|---|---|---|---|
| Knee flexors | 16.83 (2.34–20.70) (13.24 ± 7.71) | 16.71 (2.38–21.27) (13.53 ± 8.01) | 0.30 ± 1.69 |
| **Knee extensors** | **45.53 ± 14.38** | **46.53 ± 14.54**[**] | **1.01 ± 0.87**[**] |
| M. iliopsoas | 5.98 ± 3.28 | 5.40 ± 3.46 | −0.58 ± 0.86 |
| M. gluteus maximus | 28.50 ± 14.76 | 27.96 ± 14.08 | −0.54 ± 1.76 |
| M. adductor magnus | 7.87 ± 6.83 | 7.39 ± 6.29 | −0.48 ± 0.78 |
| M. adductor longus | 3.11 ± 1.98 | 2.99 ± 1.96 | −0.12 ± 0.41 |
| Sum of CSA | 104.23 ± 43.45 | 103.83 ± 42.76 | - 0.42 ± 2.78 |

Notes.

[**] $p < 0.01$.

The mean difference is calculated as follow-up value minus baseline value. Contractile cross-sectional area (CSA) is measured in $cm^2$. Values are given in mean and SD; Data without normal distribution are also presented as median and range.

**Table 7  The fat fraction of the late onset Pompe Disease patients at baseline and follow-up.**

| Fat fraction | Baseline | Follow-up | Mean difference + SD |
|---|---|---|---|
| Knee flexors | 0.21(0.06–0.75) (0.32 ± 0.27) | 0.18 (0.08–0.76) (0.32 ± 0.28) | 0.00 ± 0.02 |
| M. biceps longum | 0.15 (0.08–0.76) (0.32 ± 0.28) | 0.14 (0.10–0.76) (0.32 ± 0.29) | 0.00 ± 0.02 |
| M. semimembranosus | 0.32 ± 0.26 | 0.32 ± 0.26 | −0.00 ± 0.03 |
| M. semitendinosus | 0.18 (0.05 –0.77) (0.31 ± 0.27) | 0.19 (0.08–0.78) (0.32 ± 0.28) | 0.01 ± 0.03 |
| Knee extensors | 0.12 (0.03–0.49) (0.15 ± 0.13) | 0.13 (0.07–0.50) (0.16 ± 0.13) | 0.00 ± 0.03 |
| M. vastus med | 0.11 (0.03–0.75) (0.18 ± 0.22) | 0.13 (0.07–0.78) 0.21 ± 0.23 | 0.03 ± 0.07 |
| M. vastus lateralis et intermedius | 0.11 (0.04–0.73) (0.17 ± 0.21) | 0.10 (0.06–0.76) 0.17 ± 0.22 | 0.00 ± 0.02 |
| M. sartorius | 0.16 ± 0.06 | 0.15 ± 0.06 | −0.01 ± 0.02 |
| M. rectus femoris | 0.10 ± 0.05 | 0.09 ± 0.04 | −0.01 ± 0.03 |
| M. iliopsoas | 0.11 (0.05–0.73) (0.17 ± 0.21) | 0.08 (0.05–0.69) (0.15 ± 0.20) | −0.02 ± 0.03 |
| M. gluteus maximus | 0.32 ± 0.24 | 0.33 ± 0.24 | 0.00 ± 0.03 |
| M. adductor magnus | 0.37 ± 0.29 | 0.38 ± 0.30 | 0.01 ± 0.03 |
| M. adductor longus | 0.16 (0.02–0.75) (0.23 ± 0.22) | 0.15 (0.06–0.73) 0.22 ± 0.22 | −0.00 ± 0.04 |
| Sum of fat fractions | 0.26 ± 0.18 | 0.26 ± 0.20 | −0.00 ± 0.02 |

Notes.

The mean difference is calculated as follow-up value minus baseline value.

Values are given in mean and SD; where data were not normally distributed median and range is given first.

they only used semiquantitative strength measurements. Similar close correlations between strength and contractile CSA have also been described in other neuromuscular disorders supporting the sensitivity of the MRI method (*Morrow et al., 2016*; *Wokke et al., 2014*).

In contrast to previous studies of patients with LOPD and other neuromuscular disorders (*Pichiecchio et al., 2004*; *Morrow et al., 2016*; *Horvath et al., 2015*; *Figueroa-Bonaparte et al., 2016*; *Alejaldre et al., 2012*; *Mul et al., 2017*; *Andersen et al., 2017*; *Khan et al., 2019*; *Nuñez Peralta et al., 2020*), we did not find any correlation between fat fraction and muscle strength, with the exception of the knee flexors. In the studies by Bonaparte et al. and Khan et al. there was a very close relationship between motor function tests including muscle strength and fat fraction. The discrepancy with our study may be due to more severely affected patients and a larger study population compared to our study. Interestingly, our findings are similar to those of *Figueroa-Bonaparte et al. (2016)* who also studied muscles in patients with LOPD applying quantitative muscle strength measurements and MRI, and only found strong correlations for knee flexors ($r = 0.70$), while the correlation for knee extensors was weaker ($r = 0.63$). In line with our findings they reported no correlation for the hip flexors whereas they did not report on the hip extensors.

Unexpectedly, in our healthy control group we found only strong correlations for the contractile CSA of the knee extensors and hip flexors. This may be explained by a low number of subjects included. In healthy subjects there is a great variation in strength to CSA ratio (*Maughan, Watson & Weir, 1983*), and this may explain our findings considering the larger variability of isokinetic testing of the knee flexors and the hip flexors and extensors as compared to the knee extensors.

In our LOPD patients, the mean fat fraction did not correlate with the overall clinical status, defined from the sum-rank score of the rankings from the clinical tests (vital capacity, SF-36, manual muscle testing and 6MWT). This is not surprising since some studies show a close correlation between fat fraction and functional tests, such as the 6MWT and "time to climb up/go down 4 stairs" (*Morrow et al., 2016*; *Figueroa-Bonaparte et al., 2016*; *Fischer et al., 2016*; *Lollert et al., 2018*; *Khan et al., 2019*), while other studies did not find any correlation (*Ravaglia et al., 2010*; *Lollert et al., 2018*). Such inconsistent findings may be explained by the high clinical heterogeneity of patients with LOPD and similar diseases (*Harlaar et al., 2019*). Some patients have substantial weakness, pain (*Güngör et al., 2013*) and fatigue (*Hagemans et al., 2007*), whereas others are asymptomatic. Pain and fatigue may impair physical performance unrelated to the conditions of the muscles.

During the follow-up period the only statistically significant change observed was a 2% increase of CSA of the knee extensors. No statistically significant changes were observed for any of the clinical tests, the fat fractions or the remaining CSAs. This is in contrast to the study by Khan et al. and Nunez-Peralta et al. who found a 1.3% and 1.9% annual increase in the fat fraction in patients receiving ERT, respectively (*Khan et al., 2019*; *Nuñez Peralta et al., 2020*). In the studies, however, isokinetic dynamometry was not performed which impedes comparison between the studies.

An increase in size of the knee extensors has previously been described in treatment-naive LOPD patients: In two earlier studies by *Pichiecchio et al. (2009)* and *Ravaglia et al. (2010)* the muscle mass increased after only 6 months of ERT, and more recently a similar increase has been observed in the Embassy study (*Van der Ploeg et al., 2016*). A possible explanation is that a patient who started on ERT also started to be more active. However, this is not the

case of our population, where only 3 patients started ERT treatment in relation to study inclusion.

During the follow-up period the mean fat fraction of the muscles remained stable, which is as described in the Embassy study. However, there was a large variation in the relative changes in fat fraction during the follow-up. Similar variations in changes of fat fraction have also been observed in patients with LGMD21 (*Willis et al., 2013*). In contrast, *Carlier et al. (2015)* observed an increase in fat fraction in 23 treated and untreated LOPD patients after only one year, with more rapid muscle degeneration in patients not receiving ERT. Also, despite ERT, *Pichiecchio et al. (2009)* and *Ravaglia et al. (2010)* reported an increased fat fraction of thigh muscles over time by quantitative MRI. Compared to these studies, our population was considerably younger (mean age: 36 years, compared to 49 *Carlier et al. (2015)*, 61 *Pichiecchio et al. (2009)* and 54 *Ravaglia et al. (2010)* years, respectively). As fat replacement is part of the physiological ageing of striated muscles, this may partly explain the conflicting findings on muscle degeneration.

A recent retrospective study by *Lollert et al. (2018)* including 13 patients with LOPD (age 30 ±17.4, range 6–13) determined changes over time in intramuscular fat in the psoas major muscle and the paraspinal muscles. After 39 months, the fat fraction had increased in the psoas major muscle. However, subgroup analysis of 7 subjects, who were followed for a longer time, did not reveal any significant change in fat fraction, indicating the general challenge of receiving statistical power in evaluation of rare disorders. Interestingly, in a newly published study by *Rehmann et al. (2020)*, use of diffusion tensor imaging in MRI disclose abnormalities in muscle groups with low fat fraction which remain undetected by more conventional MRI techniques.

Our findings indicate that the contractile CSA more closely relates to the muscle strength of the given muscle as compared to fat fraction, which is in agreement with *Carlier et al. (2016)*.

Muscle imaging is a valuable tool to evaluate disease status, as it reflects muscle degeneration in a quantitative and objective way. However, due to considerable variations, muscle imaging should be considered as a supplement to clinical tests and questionnaires.

There are a number of limitations in our study. First, MR image resolution was set to 1 x 1 mm in-plane, which sets the lower boundary for precision of the analysis and thereby the sensitivity to detect small changes. However, we did use Dixon over T1 weighed images, thereby, applying a more robust method to alleviate the B0 and B1 field inhomogeneities often present in 3T MR imaging. Second, the limited number of participants does not allow us to perform subgroup analysis, e.g., drug naïve versus treated patients, and patients with low versus high levels of physical activity. Furthermore, not all patients performed all muscles tests which further lower the number of subjects in some analyses. As mentioned, the number of participants is a limiting factor, leading to indication of trends rather than showing statistical differences. The expected difference can be indicated by evaluating results from studies with 41 LOPD (13 in follow up) (*Lollert et al., 2018*) and 32 LOPD (22 ERT, 10 HyperCKemia) (*Figueroa-Bonaparte et al., 2018*) differences measured from MRI segmented muscles. Our study is also limited by a large variation in patient characteristics, including considerable differences in clinical severity

and a large age span (19–62 years), which makes subgroup analysis challenging. Due to the low prevalence of LOPD more detailed studies with stratification will only be possible in multicentre studies with standardized MRI protocols, as proposed by *Hollingsworth et al. (2012)*. This will furthermore strengthen the power of the statistical analysis and thereby providing a more precise view of muscle involvement in LOPD.

## CONCLUSION

Contractile CSA closely correlates to muscle strength in patients with LOPD. The average fat ratio of the thigh and hip muscles did not correlate to the 6MWT or the sum rank-score from conventional clinical tests. During the follow-up period the contractile CSA of the knee extensors increased by 2%, whereas, fat fraction of the hip and thigh muscles as well as functional tests remained unchanged.

### Funding
Funding was provided by Sanofi Genzyme and the ''A.P. Møller og Hustru Chastine McKinney Møllers Fond til Almene Formål''. The funders had no role in study design, data collection and analysis, decision to publish, or preparation of the manuscript.

### Grant Disclosures
The following grant information was disclosed by the authors:
A.P. Møller og Hustru Chastine McKinney Møllers Fond til Almene Formål.

### Competing Interests
Henning Andersen has received research, travel support and speaker honoraria from Octapharma, CSL Behring, Novo, Alexion and Genzyme/Sanofi, and has also served as consultant on advisory board of NMDPharma within the last 5 years.

John Vissing has received research and travel support and speaker honoraria from Genzyme/Sanofi, Ultragenyx Pharmaceuticals and aTyr Pharmaceutical, and served as consultant on advisory boards of Genzyme/Sanofi, aTyr pharmaceuticals, Ultragenyx Pharmaceuticals, Sarepta, NOVO Nordisk, Alexion Pharmaceuticals and Stealth Bio Therapeutics within the last 5 years.

Julie Schjødtz Hansen reports having received research support, honoraria, and travel funding from Sanofi Genzyme.

### Author Contributions
- Michael Vaeggemose conceived and designed the experiments, performed the experiments, analyzed the data, prepared figures and/or tables, authored or reviewed drafts of the paper, and approved the final draft.
- Rosa Andersen Mencagli performed the experiments, analyzed the data, prepared figures and/or tables, authored or reviewed drafts of the paper, and approved the final draft.

- Julie Schjødtz Hansen analyzed the data, authored or reviewed drafts of the paper, and approved the final draft.
- Bianca Dräger, Steffen Ringgaard, John Vissing and Henning Andersen conceived and designed the experiments, authored or reviewed drafts of the paper, and approved the final draft.

## Clinical Trial Ethics

The following information was supplied relating to ethical approvals (i.e., approving body and any reference numbers):

All participants gave informed signed consent and were examined at the Department of Neurology, Aarhus University Hospital, Denmark. The study was approved by the local Ethical Committee (application no. 49089) and by the Danish Data Protection Agency.

## Data Availability

Patients with POMPE are rare, and we have included all 10 patients who were diagnosed with POMPE in Denmark (the total population of Denmark is around 6.000.000) in the study. Since this study was designed, the General Data Protection Regulation (EU GDPR) has been introduced to our healthcare system and our understanding of the Danish Data Protection Agency (DDPA) leads us to believe that the MRI data cannot be published online to protect the participants' identities.

Nevertheless, representative images are included in one of the primary Figures.

## Clinical Trial Registration

The following information was supplied regarding Clinical Trial registration:

no. 49089.

## Supplemental Information

Supplemental information for this article can be found online at http://dx.doi.org/10.7717/peerj.10928#supplemental-information.

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
