# Peer review of "Function, structure and quality of striated muscles in the lower extremities in patients with late onset Pompe Disease—an MRI study"

_PeerJ, doi:10.7717/peerj.10928_

## Round 0.1 · original submission · Major Revisions

While the reviewers felt your paper was interesting, one reviewer was concerned that it added little to the field. I would recommend paying particular attention to the rationale for your study and how it addresses gaps in the research.

Reviewer 1 ·

Basic reporting

The language is clear and adequate
The literature references are comprehensive
The article is well structures and also the figures are informative
The results although well presented are not very striking

Experimental design

The study design is already proposed in different paper and do not add any novelty

Validity of the findings

The study is not quite original because several previous papers have address this issue and with a number of patients more relevant.

Additional comments

The paper is of some interest but not quite original because this issue have been raised but other studies although not all of them were univocal.
To increase the meaning I would suggest to do a review of all the studies so far reported focusing on similarities, limitations and include these data

·

Basic reporting

No comment

Experimental design

No comments

Validity of the findings

No comments

Additional comments

Very interesting article giving important information on utility of MRI in muscular involvement in Pompe disease.
A very important information is that CSA correlate well with muscular strength and could be a practical tool in order to follow therapeutics effectiveness.
I just regret that the follow up period and the time frame for enzymotherapy treatment is a little short.
I could be realy relevant to have more date over a longer period.

---

## Round 0.2 · accepted · Accept

The reviewers comments have been adequately addressed in this version.